# Recovery of Lysosomal Acidification and Autophagy Flux by Attapulgite Nanorods: Therapeutic Potential for Lysosomal Disorders

**DOI:** 10.3390/biom15050728

**Published:** 2025-05-16

**Authors:** Yuanjing Hao, Xinru Fan, Xiaodan Huang, Zhaoying Li, Zhiyuan Jing, Guilong Zhang, Yuxue Xu, Na Zhang, Pengfei Wei

**Affiliations:** School of Pharmacy, Shandong Technology Innovation Center of Molecular Targeting and Intelligent Diagnosis and Treatment, Binzhou Medical University, Yantai 264003, China; 2022081546@stu.bzmc.edu.cn (Y.H.); 2022081551@stu.bzmc.edu.cn (X.F.); huangxd@stu.bzmc.edu.cn (X.H.); 2024072029@stu.bzmc.edu.cn (Z.L.); 2022082600@stu.bzmc.edu.cn (Z.J.); glzhang@bzmc.edu.cn (G.Z.); xuyuxue@bzmc.edu.cn (Y.X.)

**Keywords:** attapulgite, lysosomes, autophagy flux, mutant huntingtin, non-alcoholic fatty liver disease, neurodegenerative diseases

## Abstract

Dysfunction of the lysosome and autophagy–lysosome pathway is closely associated with various diseases, such as neurodegenerative diseases, non-alcoholic fatty liver disease (NAFLD), etc. Additionally, chloroquine is a clinically widely used drug for treating malaria and autoimmune diseases, but long-term or high-dose administration may lead to significant toxic side effects. Attapulgite (ATT), a natural nanomaterial with excellent adsorption capacity and biocompatibility, herein demonstrated a novel biological function in regulating the lysosomal and autophagy–lysosome pathway. ATT could be effectively internalized into lysosome-related acidic compartments. Further study revealed that ATT could restore lysosomal pH, activate cathepsin D, alleviate autophagy blockage in chloroquine-treated cells, and reduce chloroquine-elicited cell death. In a cell model related to Huntington’s disease, treatment with ATT reinforced the degradation of the mutant huntingtin proteins by increasing cathepsin D maturation and autophagy flux. ATT could also promote lipid droplet clearance in hepatocytes with palmitic acid-induced steatosis, reduce hepatic lipid accumulation, and improve fasting blood glucose in high-fat-diet-induced NAFLD mice. These findings establish ATT as a lysosomal modulator, providing a foundation for its therapeutic potential in mitigating the adverse effects associated with long-term chloroquine use, especially improving neurodegenerative and metabolic disorders.

## 1. Introduction

Lysosomes are acidic organelles enclosed by a single membrane within cells, containing a variety of acidic hydrolases (such as proteases, nucleases, and glycosidases) that efficiently break down biomacromolecules, like proteins, lipids, carbohydrates, and nucleic acids [1,2]. Their membranes are embedded with key membrane proteins, such as lysosome-associated membrane protein 1 (LAMP1/CD107a) and 2 (LAMP2/CD107b), which maintain lysosomal homeostasis and functional integrity [1,3]. Additionally, the vacuolar-type ATPase (V-ATPase) on the lysosomal membrane pumps protons (H^+^) into the lysosome, maintaining its acidic environment and ensuring optimal activity of the hydrolases [4]. The lysosomal membrane also contains various important proteins, including lysosomal membrane proteins (e.g., LIMP2), ion channels (e.g., TMEM175), receptor proteins (e.g., M6PR and LAMP2A), fusion proteins (e.g., Rab7 and SNARE proteins), and signaling proteins (e.g., mTORC1) [1,3]. These proteins work synergistically to maintain the lysosome’s acidic environment, material degradation, ion balance, membrane fusion, and signal transduction, enabling it to play a central role in intracellular digestion, material recycling, and stress responses. Through processes, such as intracellular digestion, material recycling, nutrient sensing, receptor regulation, and exocytosis, lysosomes participate in cellular metabolism, waste clearance, and stress responses [5,6].

Lysosomes are closely related to the autophagy process. Autophagosomes fuse with lysosomes to form autolysosomes, which degrade damaged organelles and misfolded proteins through acidic hydrolases within the lysosomes, maintaining intracellular environmental stability [7]. Dysfunction of the lysosome and autophagy–lysosome system is closely associated with various diseases, such as neurodegenerative diseases (Alzheimer’s disease, Parkinson’s disease, Huntington’s disease), metabolic diseases (non-alcoholic fatty liver disease, diabetes), inflammation-related diseases (pancreatitis, atherosclerosis), and lysosomal storage disorders (Gaucher disease, Niemann–Pick disease) [8,9,10]. In these diseases, lysosomal and autophagy dysfunction leads to intracellular waste accumulation, metabolic disorders, and cellular damage, highlighting their central role in cellular health and disease. For instance, the prevalence of Huntington’s disease (HD) is approximately 0.4 to 4.8 affected individuals per 100,000 people [11], and the pathological mechanism of HD is closely related to dysfunction of the autophagy–lysosome pathway (ALP). The aggregation of mutant huntingtin protein (mHTT) impairs autophagy, preventing the effective clearance of toxic proteins, while the decline in lysosomal function further obstructs the degradation process, exacerbating neuronal damage [12,13,14,15]. Autophagy could clear lipid droplets, damaged mitochondria, and toxic proteins in hepatocytes, maintaining liver homeostasis. However, in NAFLD, impaired autophagy leads to lipid accumulation and exacerbated inflammation. Factors such as a high-fat diet and oxidative stress suppress autophagy, promoting disease progression [16,17]. The global prevalence of NAFLD in adults is around 30% [18]. NAFLD, as a representative metabolic disease, demonstrates a vicious cycle between lipid metabolism dysregulation and lysosomal functional impairment [16,19]. Consequently, restoring lysosomal acidification and autophagic flux has the potential to reverse NAFLD and represents a novel treatment strategy.

Additionally, chloroquine (CQ) is a drug widely used in clinical settings for treating malaria and autoimmune diseases (such as rheumatoid arthritis and systemic lupus erythematosus) and for antiviral therapy [20]. However, long-term or high-dose use of CQ may cause significant toxic side effects, including retinopathy, arrhythmias, neuromuscular toxicity, gastrointestinal reactions, hematological abnormalities, and skin reactions [21,22]. These side effects primarily stem from its mechanisms of inhibiting lysosomal function, disrupting mitochondrial metabolism, and affecting ion channels [22,23,24]. Consequently, the development of novel nanomaterials with robust lysosomal regulatory capabilities and superior biosafety has emerged as a prominent area of investigation within the biomedical field.

ATT, a naturally occurring one-dimensional nanosilicate material, possesses remarkable ion exchange properties and biocompatibility due to its distinctive layered chain structure, controllable surface charge, and abundant hydroxyl groups. In 2012, ATT was approved as a food additive for use as a processing aid in China due to its stability and safety (GB 29225-2012, National food Safety Standard of China). Recent studies have highlighted its unique advantages in biomedical applications, including drug delivery and antimicrobial dressings [25,26,27]. Recent studies have also identified independent anti-inflammatory properties of ATT [28,29], expanding its clinical application prospects. However, its potential regulatory effects on organelle microenvironments and autophagy pathways remain inadequately explored. Notably, the abundant MG-aluminosilicate structures present on the surface of ATT may influence lysosomal pH through proton buffering, while their nanoscale dimensions (20~50 nm in diameter) and surface-active groups may further contribute to the modulation of autophagy-related signaling pathways. These characteristics indicate that ATT holds significant promise for applications in the intervention of lysosome-related diseases [30,31,32].

The current investigation aims to elucidate the regulatory mechanisms by which ATT influences lysosomal homeostasis and autophagy pathways while also systematically assessing their therapeutic potential in models of neurodegenerative and metabolic diseases. Employing a comprehensive array of research methodologies, including molecular biology, cellular imaging, and animal models, this study represents the first instance that demonstrates how these nanoparticles facilitate the clearance of toxic protein aggregates and the restoration of lipid metabolism disorders. This is achieved through the reconstruction of the lysosomal acidic microenvironment and the activation of autophagic degradation pathways. The findings of this research not only broaden the application scope of natural nanomaterials in organelle engineering but also offer innovative approaches for the development of multi-target therapeutic strategies based on the regulation of autophagy.

## 2. Materials and Methods

### 2.1. Reagents and Antibodies

Phosphate buffer salt solution (P1003), penicillin–streptomycin (P1400), Hoechst33342 (C0031), proteasome inhibitor (IKM1010), 5 × loading buffer (20317ES05), 1.5 M Tris-HCl buffer (pH8.8; T1010), 1 M Tris-HCl buffer (pH6.8; T1020), Bovine Serum albumin (A8010), and an Oil Red O staining kit (G1261) were purchased from Beijing Solaibao Technology Co., LTD., Beijing, China. DMEM (PM150225) and RPMI-1640 (PM150110) medium were purchased from Wuhan Punuo Life Technology Co., LTD., Wuhan, China. Lysosomal red fluorescent probe (C1046), Western and IP cell lysate (P0013), a CCK-8 cell counting kit (C0037), and hydrochloric acid ethanol rapid differentiation solution (C0163) were purchased from Shanghai Biyuntian Biotechnology Co., LTD., Shanghai, China. Chloroquine diphosphate (S6999) and wortmannin (S2758) were purchased from Selleck, Houston, TX, USA. Palmitic acid (57-10-3) was purchased from Shanghai McLean Co., Ltd., Shanghai, China. Bafilomycin A1 (HY-100558) was purchased from Med-ChemExpress, Monmouth Junction, NJ, USA. Laboratory rat maintenance feed (for the normal-fat diet) was purchased from Jiangsu Sypharma Bioengineering Co., Ltd., Nanjing, China. mRFP-GFP-LC3-expressing virus was purchased from Applied Biological Materials Inc., Canada (ABM) Company, Richmond, BC, Canada. High-fat rat feed was purchased from ResearchDiets, New Brunswick, NJ, USA. Attapulgite was purchased from Mingguang Feizhou New Materials Co., Ltd., Mingguang, China. An alanine aminotransferase (alanine aminotransferase/ALT/GPT) test box (Lai’s method), Microplate method (C-009-2-1), was purchased from Nanjing Jiancheng Bioengineering Research Institute Co., Ltd., Nanjing, China. A GOT/AST kit (Enzyme labeling method) (C010-2-1) was purchased from Nanjing Jiancheng Bioengineering Research Institute Co., Ltd., Nanjing, China. A lysosomal yellow/blue fluorescent probe (40768ES50) was purchased from Shanghai Yisheng Biotechnology Co., Ltd., Shanghai, China.

Cathepsin D antibody (21327-1-AP) was purchased from Proteintech Group, Wuhan, China. GAPDH antibody (AF7021) and SQSTM1/p62 antibody (AF5384) were purchased from Affinity Biosciences, Cincinnati, OH, USA. LC3B antibody (NB100-2220) was purchased from Novus Biologicals, Centennial, CO, USA. GFP antibody (AE078) was purchased from ABclonal Technology, Woburn, MA, USA.

### 2.2. Preparations of ATT

To prepare a 10 mg/mL solution of ATT, 100 mg of the ATT powder was weighed, and three portions of steaming water were added to dissolve the powder and reach a constant volume of 10 mL. An ultrasonic solution of ATT nanorods was obtained in an ultrasonic cleaning machine for 2 h and stored at 4 °C for further use. ATT was characterized using the following instruments: the Zetasizer Nano ZS-90 from Malvern Panalytical, Malvern, UK; the transmission electron microscope JEM-1400 from JEOL, Tokyo, Japan; and the XRD instrument Rigaku Smartlab SE from Japan.

### 2.3. Preparations of FITC-ATT

To create a 1 mg/mL solution of FITC-ATT, 10 mg of fluorescein isothiocyanate powder was weighed out and dissolved in 10 mL of dimethyl sulfoxide. A 10 mg/mL solution of ATT was prepared and mixed with the 1 mg/mL FITC solution in a 1:1 mass ratio. The mixture was stirred at room temperature overnight and then centrifuged three times at 10,000 rpm for 10 min each time. After centrifugation, the mixture was re-dispersed in three portions of steaming water to obtain FITC-labeled ATT, which was stored at 4 °C for future use. The successful labeling was confirmed by fluorescence imaging using a multifunctional fluorescence imaging system (Azure 600, Azure Biosystems, Dublin, CA, USA).

### 2.4. Construction of mRFP-GFP-LC3/HeLa Cells

HeLa cells were seeded into 24-well plates; then, mRFP-GFP-LC3 lentivirus (Applied Biological Materials Inc., Richmond, BC, Canada, CS5005498) and polybrene were added to the cell culture medium. After 24 h of incubation, the medium was replaced with a fresh culture medium for another 48 h of incubation. Then, viral infection efficiency was assessed under a fluorescence microscope. Once infection was successful, puromycin at 1 μg/mL was added for selection.

### 2.5. Cell Culture

GFP-LC3/HeLa cells and GFP-Htt(Q74)/PC12 cells were kindly gifted from professor Longping Wen’s group [33,34]. HeLa cells, EA.hy926 cells, HaCaT cells, HepG2 cells, GFP-Htt(Q74)/PC12 cells, GFP-LC3/HeLa cells, and mRFP-GFP-LC3/HeLa cells were grown in a DMEM high-glucose medium supplemented with 10% fetal bovine serum and 1% penicillin–streptomycin solution. All cell lines were kept in an incubator at 37 °C with 5% CO_2_.

### 2.6. Flow Cytometric Analysis of the Internalized FITC-Labeled ATT

Hela cells were placed on a six-well plate, and 500 μg/mL of FITC-labeled ATT was added to the well plate. After incubation for different times, the material was digested and centrifuged with a pancreatic enzyme and then placed on ice for flow cytometry.

### 2.7. Observation of ATT by Biological TEM (Bio-TEM)

HeLa cells were plated in cell culture dishes and allowed to adhere for 24 h. Subsequently, they were exposed to 500 μg/mL of ATT nanorods for 6 h. After treatment, the cells were gently rinsed with phosphate-buffered saline (PBS) and fixed with 2.5% glutaraldehyde at 4 °C for 2 h. Following fixation, the cells were dehydrated using a graded ethanol series and embedded in epoxy resin. The embedded samples were then sectioned into ultra-thin slices and stained for visualization. Finally, all samples were examined using a JEM-1400 transmission electron microscope (JEOL Ltd., Tokyo, Japan) at an acceleration voltage of 100 kV.

### 2.8. Western Blotting

Cells were collected and cracked in RIPA lysate or WB/IP lysate containing a protease inhibitor (Beyotime, Haimen, China, P0013). The protein concentration was determined by the Bradford method and mixed with a 5× protein loading buffer. The cells were immersed in a metal bath at 99 °C for 10 min. A methanol-activated PVDF membrane (Millipore, Darmstadt, Germany, ISEQ00010) was used for protein transfer. First, 5% skim milk was incubated at room temperature for 1 h, followed by incubation with a pre-configured primary antibody at 4 °C overnight and then incubation with a secondary antibody at room temperature for 1 h. The proteins of interest were visualized by an enhanced chemiluminescence system (SCG-W2000, Servicebio, Wuhan, China).

### 2.9. Cell Viability

Cells were grown in 96-well plates at a density of about 10,000 cells per well. After the specified treatment, CCK-8–cell medium (1:9, 100 μL) was added and incubated for 45 min. Cell viability was measured with an enzymoleter, and signals were detected at 450 nm.

### 2.10. Palmitate: BSA Preparation

Palmitate was first dissolved in DMSO and subsequently dissolved in DMEM medium (glucose-free) containing 6.7% fatty-acid-free BSA at 45 °C to prepare a 4 mM (10×) reserve solution. For control BSA conditions, a 10× DMEM medium reserve containing 5% BSA and 1% DMSO was used. For treatment conditions, a 10× reserve solution was added to the DMEM medium containing 5% FBS, 50 U/mL penicillin, and 50 g/mL streptomycin. A 0.45 μm syringe filter was used for aseptic filtration of the adjusted medium.

### 2.11. Nile Red Staining

To observe the lipid droplets by immunofluorescence, the differentially treated cell samples were fixed in 4% paraformaldehyde for 30 min and stained with Nile red in the dark at room temperature for 15 min. Then, after cleaning the medium, images were observed under a fluorescence microscope and captured.

### 2.12. Construction and Treatment of the NAFLD Model

Male 4-week-old C57BL/6J mice were purchased from Hangzhou Ziyuan Laboratory Animal Technology Co., LTD., Hangzhou, China, The animal research protocol has been approved by the Binzhou Medical University Animal Ethics Committee (2020-409, 19 December 2020). After a week of adaptive feeding, the C57BL/6J mice were randomly divided into four groups. Two groups were fed a normal diet (ND, 1010088, Jiangsu Synergetics Biological Engineering Co., LTD., Nanjing, China) for 16 weeks, while the other two groups were fed a high-fat diet (HFD, D12492, 60% kcal from fat, American Research Diets) for the same duration. Subsequently, the two groups on the ND received intravenous injections of 10 mg/kg ATT or PBS every other day, four times, and the two groups on the HFD were also given intravenous injections of 10 mg/kg ATT or PBS every other day, four times. The groups were named as follows: the ND-PBS group (n = 3), the ND-ATT group (n = 3), the HFD-PBS group (n = 5), and the HFD-ATT group (n = 7). Finally, under anesthesia, blood samples were obtained from the mice, followed by euthanasia to facilitate the collection of tissue samples. Subsequently, the weight of the liver was measured, and biochemical indexes were assessed. There were no exclusions.

### 2.13. Oral Glucose-Tolerance Test (OGTT)

The *OGTT* was carried out on mice that were fasted overnight for 12 h. After determination of fasted blood glucose levels, each mouse was administered 2 g/kg body weight of glucose by intragastric administration. Blood glucose levels were detected from the tail vein after 15, 30, 60, 90, and 120 min. The blood glucose concentration was plotted at different times, and the area under the curve was calculated. (AUC).

### 2.14. Detection of Serum Biochemical Indexes

Venous blood was collected from the eyes of the mice after anesthesia. Whole blood was left at 4 °C for more than 2 h and then centrifuged at 3000 rpm for 20 min. The upper serum was absorbed, and ALT and AST levels in the blood were detected with alanine aminotransferase and aspartate aminotransferase (AST) test boxes.

### 2.15. Hematoxylin–Eosin Staining

The heart, liver, spleen, lungs, and kidneys were quickly extracted. Then, the blood was cleaned and placed in a tissue embedding frame and fixed in 4% paraformaldehyde for 48 h. After fixation, the tissue blocks were embedded with an automatic paraffin embedding machine. The embedded tissue was cut into 5 μm slices with an automatic paraffin microtome and attached to the pathological slides. The slides were baked at 65 °C for 2 h before dyeing. Then, dewaxing, hydration, dyeing, and sealing were carried out in sequence. A fluorescent microscope was used to observe and take images.

### 2.16. Liver Oil Red O Staining

Fresh liver tissues were taken, put in OCT embedding agent, and frozen on the surface of liquid nitrogen. The tissues were cut into slices with a thickness of 10 μm using a cold microtome and dried at room temperature for 20 min. After dyeing, the tissues were stained with an Oil Red O staining kit. After dyeing, the slices were sealed, observed, and photographed with a fluorescence microscope.

### 2.17. Cell Lysosomal Acid Detection

Cells were inoculated in a confocal dish (cell density 1 × 10^5^ cells/well), adhered to the wall for 24 h, and treated with different materials. The medium was removed, cleaned with PBS, incubated with 75 nM Lyso-Tracker Red preheated at 37 °C for 30 min, and cleaned with PBS; then, a new complete medium was added. The cells were observed, and images were taken using a laser scanning confocal microscope.

### 2.18. Statistical Analysis

Data were expressed as the mean ± standard error of the mean (SEM). All analyses were performed using GraphPad Prism 8.0 (GraphPad Software, San Diego, CA, USA). Statistical significance was calculated using two-tailed Student’s t-tests for comparisons between two groups. * *p* < 0.05, ** *p* < 0.01, and *** *p* < 0.001 was considered statistically significant. All mice were randomized for treatment.

## 3. Results

### 3.1. Characterizations of ATT

Initially, we performed characterization of the ATT. The results obtained from transmission electron microscopy (TEM) (Figure 1A) revealed that ATT exhibits a well-dispersed morphology characterized by a needle–rod structure. The measurements conducted using Malvin’s nanoparticle size analyzer indicated that the hydrated particle size of ATT is 396.0 ± 11.0 nm (Figure 1B), with a Zeta potential of −16.6 ± 0.3 mV. X-ray diffraction (XRD) analysis (Figure 1C) showed that the 2θ angle of ATT displays characteristic peaks that corresponded to the fundamental framework of ATT at 8.2°, as well as internal silicon–oxygen absorption peaks at 13.6°, 19.8°, and 20.9°. The presence of characteristic absorption peaks for quartz, at 26.7°, and montmorillonite, at 35.2°, further indicated the presence of impurities from these minerals. Fourier transform infrared spectroscopy (Figure 1D) identified characteristic peaks associated with the magnesia hexahedron within ATT at 459.2 cm^−1^, 511.8 cm^−1^, and 883.4 cm^−1^. Additionally, the peaks at 986.7 cm^−1^ and 1033.5 cm^−1^ correspond to the stretching vibrations resulting from the asymmetry of the silicon–oxygen structure, while the peak at 1211.0 cm^−1^ is attributed to the stretching vibrations of the silicon–oxygen–silicon structure. Collectively, these physical and chemical analyses suggest that ATT exhibits favorable dispersion and possesses a crystal structure and composition that align with standard reference materials.

### 3.2. Cellular Uptake and Intracellular Localization of ATT

Further investigations revealed that ATT could be effectively internalized into lysosome-related acidic compartments. First, ATT was labeled with fluorescein isothiocyanate (FITC) for fluorescence tracking (Figure 2A). Flow cytometric analysis indicated that FITC-labeled ATT could be internalized by HeLa cells with a time-dependent pattern (Figure 2B). Furthermore, bio-TEM imaging analysis of chemically fixed cell samples (Figure 2C) clearly demonstrated the presence of ATT with distinct morphological characteristics within the vesicular structures of HeLa cells, confirming its cellular internalization. To precisely determine the subcellular localization of internalized ATT, we performed confocal laser scanning microscopy (CLSM) analysis using a triple-labeling approach. ATT was labeled with fluorescein isothiocyanate (FITC) for fluorescence tracking, while the cellular components were differentially stained with Hoechst 33,342, for nuclei visualization, and Lyso-Tracker Red, a specific fluorescent probe that selectively accumulates in acidic lysosomal compartments, for live-cell imaging [35]. CLSM analysis revealed a significant colocalization of FITC-labeled ATT with Lyso-Tracker Red-stained acidic organelles post-internalization (Figure 2D–F), providing conclusive evidence that the internalized ATT could be localized within lysosomal compartments.

### 3.3. Restoration of Lysosomal Acidity and CTSD Maturation by ATT

CQ has the capacity to alkalinize lysosomes, which leads to the inactivation of acidic hydrolases within these organelles and disrupts the fusion of autophagosomes with lysosomes during the later stages of autophagy, thereby impeding autophagic flux [36,37]. Hence, CQ-treated cell models were deployed to investigate whether ATT could restore lysosomal acidity and the maturation of acid hydrolases. With reference to the earlier literature [35,38], Lyso-Tracker Red was employed to probe acidic lysosome-related organelles. As shown in Figure 3A, decreased Lyso-Tracker Red staining by CQ was rescued by ATT treatment, suggesting ATT possessed the capacity to restore lysosomal acidification. Meanwhile, the cells treated with ATT alone also increased Lyso-Tracker Red staining, indicating that ATT could also induce an effect of lysosome acidification in the normally cultured cells. Consequently, it could be concluded that ATT has the potential to restore the acidity of lysosomes that have been alkalinized by CQ.

Cathepsin D (CTSD), a lysosomal aspartic protease, originates as a 52 kDa precursor termed preprocathepsin D. This precursor undergoes intramolecular autocatalysis under acidic conditions to transform into its mature 28 kDa form [39,40]. Our findings revealed that ATT alone did not significantly alter CTSD maturation in HeLa cells (Figure 3B). However, it effectively counteracted the inhibitory influence of CQ on this process. While CQ substantially impeded CTSD maturation, ATT successfully restored the expression levels of mature CTSD, indicating its ability to reacidify CQ-alkalized lysosomes and promote lysosomal enzyme maturation (Figure 3B). Moreover, our study demonstrated that ATT reactivated CTSD maturation in a dose-dependent manner in CQ-treated cells. The enhancement in CTSD maturation by ATT followed a distinct dose–response relationship, with maturation levels escalating progressively and reaching their zenith at an ATT concentration of 500 μg/mL (Figure 3C). These results underscore ATT’s capacity to mitigate CQ-induced impairment of lysosomal acid hydrolase CTSD maturation, with a pronounced dose-dependent efficacy.

### 3.4. Restoring CQ-Blocked Autophagic Flux and Reducing CQ-Induced Cytotoxicity by ATT

Autophagic flux refers to the complete dynamic process of autophagy from initiation to degradation, encompassing the formation of autophagosomes, the fusion of autophagosomes with lysosomes, and the degradation and recycling of autophagic substrates [34]. Autophagic flux is a critical indicator for assessing the functional integrity of autophagy, reflecting the cell’s ability to clear damaged organelles, misfolded proteins, and pathogens through the autophagy pathway. Given that ATT has the ability to restore the acidity of lysosomes alkalized by CQ, we proceeded to investigate whether ATT could reinstate the autophagic flux and facilitate the degradation of autophagic substrate proteins. Microtubule-associated protein 1 light chain 3 (MAP1LC3, LC3) exists in two forms in mammalian cells: the cytoplasmic form, LC3-I, and the membrane-bound form, LC3-II, which is generated through lipidation of LC3-I during autophagy induction. LC3-II specifically associates with both the outer and inner membranes of autophagosomes [33,41]. Sequestosome 1 (SQSTM1, commonly referred to as p62) is a multifunctional adaptor protein that regulates ubiquitin-mediated protein degradation and serves as a selective autophagy substrate. CQ, a known lysosomotropic agent, exerts its effects by alkalizing lysosomal pH, inactivating lysosomal acid hydrolases, and inhibiting autophagosome–lysosome fusion, consequently leading to the accumulation of autophagic substrates [42]. In our experiments, we demonstrated that CQ treatment significantly blocked autophagic flux, as evidenced by the accumulation of both LC3-II and p62 proteins (Figure 4A). Notably, this CQ-induced autophagic inhibition was effectively attenuated by co-treatment with ATT. Intriguingly, ATT alone did not exhibit significant effects on basal autophagy levels (Figure 4A). To further investigate autophagic activity, we employed GFP-LC3 reporter cells, where the formation of GFP-LC3 puncta serves as a sensitive indicator of autophagosome formation and autophagic flux. As shown in Figure 4B, ATT significantly reduced the CQ-induced accumulation of GFP-LC3 puncta, suggesting its potential role in restoring the degradation of autophagy substrates that are otherwise blocked by CQ. Consistent with these observations, the Western blot analysis revealed a dose-dependent decrease in the accumulation of LC3-II and p62 proteins following co-treatment with CQ and ATT, with higher concentrations of ATT leading to more pronounced reductions in these autophagic markers (Figure 4C,D).

Given the substantial presence of magnesium silicate, aluminum, and other constituents in ATT, we conducted a further examination to determine whether silicates, including magnesium silicate, aluminum silicate, and sodium silicate, could similarly restore the autophagic flux that is inhibited by CQ. The findings from the Western blot analysis (Figure 4E) revealed that these silicates did not mitigate the accumulation of autophagic substrate proteins that CQ inhibits. We proposed that the distinctive structure or composition of ATT might play a significant role in its ability to restore autophagic flux, highlighting its chemically specific composition for this function. Bafilomycin A1, a macrolide antibiotic, is a specific and reversible inhibitor of H+−ATPase (V-ATPase). It can block the fusion of autophagosomes and lysosomes, thereby inhibiting the acidification and protein degradation in cellular lysosomes [43]. Ammonium chloride inhibits autophagy by alkalizing lysosomes, suppressing lysosomal enzyme activity, and disrupting the fusion of autophagosomes with lysosomes [44]. Subsequently, we employed bafilomycin A1 and ammonium chloride to assess the general applicability of ATT in restoring autophagic flux. The results from the Western blotting (Appendix A) indicated that both bafilomycin A1 and ammonium chloride led to the accumulation of autophagic substrate proteins and impeded autophagic flux. However, the addition of ATT resulted in a reduction in the accumulation of these substrate proteins. This suggests that ATT could alleviate the autophagic blockage induced by bafilomycin A1 and ammonium chloride, thereby restoring autophagic flux and facilitating the degradation of autophagic substrate proteins. In conclusion, ATT possessed the capability to restore autophagic flux that is obstructed by various lysosomal inhibitors.

To further investigate the effects of ATT on autophagic flux, we employed the mRFP-GFP-LC3/HeLa cell line as a sensitive reporter system. This dual-fluorescence system takes advantage of the differential pH sensitivity of GFP (acid-labile) and mRFP (acid-stable) to monitor autophagic progression. During normal autophagic flux, the acidic environment of lysosomes quenches GFP fluorescence upon autophagosome–lysosome fusion, leaving only mRFP fluorescence detectable. However, when autophagosome–lysosome fusion is impaired, both GFP and mRFP signals persist, resulting in yellow fluorescence from their colocalization. This system enables the real-time monitoring of autophagic flux dynamics in living cells. Our experimental results demonstrated that CQ treatment significantly inhibited autophagosome–lysosome fusion, as evidenced by the persistence of the GFP signal and increased yellow fluorescence (Figure 4F,G). Notably, co-treatment with ATT effectively restored lysosomal acidity and promoted autophagosome–lysosome fusion, leading to GFP quenching and reduced yellow fluorescence. Quantitative analysis of the yellow puncta revealed that while CQ treatment significantly increased yellow fluorescence due to impaired autophagic flux, the concurrent administration of ATT substantially decreased yellow fluorescence intensity. These findings strongly suggested that ATT could counteract CQ-induced autophagic inhibition by restoring lysosomal function and facilitating autophagosome–lysosome fusion, thereby re-establishing normal autophagic flux.

It is noteworthy that CQ treatment can lead to lysosomal alkalization, which subsequently increases lysosomal membrane permeability (LMP) and induces cytotoxic effects [36]. Herein, we also validated the cytotoxic potential of CQ. The experimental data revealed that in the HeLa cells, 50 μM CQ induced only minimal cytotoxicity, whereas the 100 μM treatment resulted in significant cytotoxic effects (Appendix A). Based on these findings, the 100 μM concentration was selected for subsequent evaluation of ATT’s potential cytoprotective properties. Furthermore, we examined the potential of ATT to restore cell viability across various cell types. As shown in Figure 4H, the application of ATT alone, irrespective of its concentration, exhibited negligible effects on the viability of the aforementioned cell lines. However, when various concentrations of ATT were co-incubated with CQ for a duration of 24 h, a reduction in CQ-induced HeLa cytotoxicity was observed (Figure 4H). Following this, we assessed the impact of ATT on the viability of EA.hy926 and HaCaT cells in the presence of CQ. The results indicated that ATT could mitigate the accumulation of the autophagy substrate protein LC3-II and cytotoxicity induced by CQ in EA.hy926 and HaCaT cells (Appendix A). This protective effect may be attributed to the capacity of ATT to restore the acidity of lysosomes that had been alkalinized by CQ, which is correlated with a decrease in cell death. These findings further substantiate the protective role of ATT in the recovery of lysosomal acidity within cells.

### 3.5. Increasing Lysosomal Acidity and Accelerating the Clearance of Mutant Huntingtin by ATT

mHTT exhibits an age-dependent accumulation within the central nervous system, resulting in neuronal damage and subsequent cell death, which contributes to the pathogenesis of HD. The degradation of these misfolded, aggregation-prone proteins is primarily facilitated through the activation of autophagy [45]. Previous research has demonstrated that toxic protein aggregates can be effectively degraded through autophagy, which can be induced by small molecules and nanomaterials [46]. In this study, we examined the degradation of mHTT utilizing ATT. Our findings revealed that the addition of ATT resulted in a concentration-dependent increase in lysosomal activity within the GFP-Htt(Q74)/PC12 cells. The Western blot analysis indicated that treatment with ATT facilitated the maturation of CTSD, as evidenced by the conversion of immature CTSD to its mature form (Figure 5A), thereby suggesting that these nanorods enhance the maturation of lysosomal enzymes. Meanwhile, the GFP-Htt(Q74)/PC12 cells treated with ATT also increased Lysotracker Red staining, indicating that ATT could also increase lysosome acidification (Figure 5B,C). Furthermore, ATT was found to induce autophagy, as indicated by the conversion of LC3 protein from its LC3-I form to LC3-II and the degradation of p62 protein (Figure 5D). The co-incubation of GFP-Htt(Q74)/PC12 cells with ATT resulted in a significant promotion of mHTT degradation, which was observed to increase with higher concentrations of ATT (Figure 5E,F). To elucidate the mechanism by which ATT facilitated the degradation of mHTT, we employed the autophagy inhibitor wortmannin. The results demonstrated that wortmannin effectively inhibited the degradation of mHTT mediated by ATT, thereby confirming that the degradation occurs via the autophagy pathway (Figure 5G). Additionally, we observed that ATT enhanced the viability of GFP-Htt(Q74)/PC12 cells (Figure 5H), indicating a protective effect against toxic protein accumulation. In conclusion, ATT is capable of restoring lysosomal acidity, inducing autophagy, accelerating the lysosomal degradation of mHTT, and enhancing cell viability.

### 3.6. ATT Reacidifies Lysosomes and Restores Autophagic Flux Under Lipotoxicity

High levels of free fatty acids in the liver can impair lysosomal acidification and inhibit autophagic flux [47]. Therefore, we initially investigated in vitro experiments to assess whether ATT could mitigate the decrease in lysosomal acidity and the suppression of autophagic flux induced by palmitic acid (PA), a long-chain fatty acid, in lipotoxic cells. HepG2 cells were subjected to treatments with bovine serum albumin (BSA) as a control, PA (complexed with BSA at a 4:1 ratio), ATT + BSA, or a combination of PA and ATT (Figure 6A). The observations made through confocal laser scanning microscopy (CLSM) revealed that the cells treated with PA exhibited a significant reduction in fluorescence intensity compared to those treated with BSA, indicating a decrease in lysosomal acidity. However, the addition of ATT resulted in a restoration of fluorescence intensity in the cells, suggesting that these nanoparticles effectively counteracted the decrease in lysosomal acidity induced by PA (Figure 6B,C).

To accurately assess autophagic flux, we measured the expression levels of autophagic substrate proteins in HepG2 cells utilizing Western blot analysis, specifically targeting LC3-II and p62 (Figure 6D). The results indicated that treatment with PA significantly impaired autophagic flux in the HepG2 cells, leading to the inhibition of autophagic substrate protein degradation and subsequent accumulation of these proteins (Figure 6E). Conversely, the addition of ATT facilitated the restoration of autophagic flux and enhanced the degradation of autophagic substrate proteins (Figure 6D,E), with this effect exhibiting a dose-dependent relationship (Figure 6F,G). These findings validate the conclusion that ATT effectively reverses the inhibition of autophagic flux induced by PA by restoring lysosomal acidification.

Given the established relationship between autophagy and lipid clearance in the liver via lipophagy, we undertook in vitro investigations to assess the efficacy of ATT in mitigating lipid accumulation in hepatocytes. Utilizing CLSM for our observations, we observed a significant increase in the number of lipid droplets within the HepG2 cells following exposure to PA. Conversely, the HepG2 cells treated with ATT exhibited a notable reduction in lipid droplet accumulation (Figure 6H,I). These findings indicate that ATT reverses the decrease in lysosomal acidity induced by lipotoxicity, restores normal autophagic flux, and effectively reduces lipid accumulation.

### 3.7. ATT Alleviates the Pathological Condition in High-Fat-Diet-Induced NAFLD Mice

The high-fat diet (HFD)-induced NAFLD model is widely utilized in contemporary research as an animal model [48,49]. Prolonged feeding duration results in an increase in body weight, along with heightened blood glucose levels and the accumulation of lipids in the liver. These manifestations closely resemble the biochemical alterations and pathological histological characteristics observed in human NAFLD. Consequently, we investigated the therapeutic potential of ATT within the HFD model.

Based on the significant therapeutic efficacy of ATT on lipid-toxic HepG2 cells, we further investigated the in vivo therapeutic potential of ATT using an HFD-induced NAFLD mouse model. In this study, C57BL/6 mice were subjected to an HFD for a duration of 16 weeks. Subsequently, the mice were treated in accordance with the protocol illustrated in Figure 7A. During the experiment, we continuously monitored the weight fluctuations of the NAFLD mice and observed that the intravenous administration of ATT significantly reduced the increases in body weight (Figure 7B) and liver weight (Figure 7C) in the HFD-fed mice. Furthermore, we evaluated the alanine aminotransferase (ALT) levels and aspartate aminotransferase (AST) levels in the hepatic tissue of the HFD-fed mice (Figure 7D,E) and found that treatment with ATT led to a significant improvement in ALT levels and AST levels, which approached normal physiological ranges. These results collectively suggest that ATT is effective in reducing both body and liver weight in mice, as well as in normalizing liver function indicators, such as ALT and AST.

In the subsequent analysis, we evaluated the impact of ATT on systemic glucose regulation through the implementation of an oral glucose tolerance test (OGTT). The results indicated that treatment with ATT resulted in enhanced glucose tolerance in the mice subjected to the HFD, with their blood glucose levels decreasing to values comparable to those observed in the normal mice (Figure 7F). To further clarify the variations in the baseline fasting blood glucose levels across the different treatment groups, we computed the area under the OGTT curves. In comparison to the HFD control group, the administration of ATT led to a significant reduction in the total glucose excursion (Figure 7G), indicating a notable improvement in glucose reactivity.

To thoroughly examine whether the observed reduction in liver weight is linked to a decrease in lipid accumulation, we employed Oil Red O staining to evaluate lipid deposition in liver tissues. As illustrated in Figure 7H, the mice fed with the HFD exhibited significant lipid accumulation within the liver. In contrast, treatment with ATT resulted in a marked reduction in lipid accumulation in the hepatic tissue of these mice. Furthermore, we applied hematoxylin–eosin staining (H&E) to assess the extent of hepatic steatosis, as depicted in Figure 7H. In the control group receiving the HFD, the liver tissue sections exhibited prominent areas of hepatocyte hypertrophy, indicative of severe hepatic steatosis. However, this condition was reversed following treatment with ATT.

Furthermore, we evaluated the in vivo biosafety of the treatment and conducted H&E staining on critical tissue sections, as illustrated in Appendix A. The findings revealed no significant damage to the major organs, thereby suggesting that ATT exhibits favorable biosafety.

## 4. Discussion

Dysfunction in the lysosomal acidification and autophagy–lysosomal system significantly influences the pathogenesis and progression of various conditions, including neurodegenerative diseases, metabolic disorders, and inflammatory diseases. Consequently, the restoration of lysosomal acidity or an enhancement in autophagy flux functionality has emerged as a critical therapeutic target in disease management [50]. This study elucidates the biological effects of ATT nanorods within cells and their potential therapeutic applications. The impact of ATT on lysosomal acidification dysfunction and the autophagy–lysosomal system is presented in Figure 8. ATT nanorods can be internalized by cells and localized to lysosome-associated acidic compartments, effectively restoring the acidic environment and autophagy flux of CQ-alkalized lysosomes, thereby accelerating the degradation of autophagy substrate proteins and significantly alleviating CQ-induced cytotoxicity. In HD-related cell models, ATT nanorods enhance the activity of the lysosomal acid hydrolase CTSD, promote autophagy flux, and subsequently accelerate the degradation of mTT. Furthermore, in a PA-induced lipid accumulation model using human HepG2 cells, ATT nanorods significantly restore lysosomal acidity and autophagy flux. Notably, in an HFD mouse model, ATT nanorods demonstrate significant body weight regulation and effectively reduce hepatic lipid droplet accumulation. These findings provide important experimental evidence for the potential therapeutic applications of ATT nanorods in metabolic and neurodegenerative diseases.

This study reveals the novel regulatory effects of ATT on subcellular organelle functions. As a natural nanosilicate mineral, ATT demonstrates excellent biocompatibility, drug-loading capacity, and modifiability due to its unique nanorod structure and surface properties, showing broad application prospects in biomedical fields [51]. This research found that the promotion of reacidification of alkalized lysosomes represents a distinctive biological function of ATT: it can not only effectively reverse CQ-induced lysosomal alkalization but can also counteract autophagy inhibition caused by bafilomycin A1 and ammonium chloride. Notably, this functionality has not been observed in other silicate materials like magnesium/aluminum/sodium silicate. ATT differs from these silicates not only in its composition but also in its unique structure and properties. It has a one-dimensional-chain-layered nanostructure with surface hydroxyl groups and exchangeable cations, whereas magnesium/aluminum silicates have rigid layered structures with inert Si-O bonds [52,53], and sodium silicate is a soluble amorphous salt lacking porosity. These structural or compositional differences likely explain ATT’s distinct pH-regulating ability in lysosomes. More importantly, ATT exhibits broad-spectrum lysosomal repair capabilities, showing significant improvement effects on various pathological lysosomal dysfunctions. The current study demonstrated ATT’s distinctive lysosomal repair capability, which appears to be associated with its unique chemical composition; however, the structure–activity relationships between its physical properties (e.g., particle size, morphology) and biological functions remain to be further elucidated. Future studies will provide important theoretical foundations and technical support for developing novel nanodrugs based on precise lysosomal regulation.

The reacidification effect of ATT on disordered lysosomes may not depend on its intrinsic pH-regulating capacity. Studies have shown that nanomaterials tend to localize in lysosome-associated acidic compartments after cellular internalization. For instance, pH-responsive materials, such as PLGA- or PLA-based nanoparticles [19,35,54,55], can degrade within lysosomes and release acidic groups, thereby restoring lysosomal pH. However, our in vitro experiments demonstrated that ATT exhibited no significant pH-modulating capacity under either neutral or mildly acidic conditions (Appendix A), which might be attributed to its excellent acid stability. Notably, various inorganic nanomaterials, including gold nanoparticles [56], cerium oxide nanoparticles [57], graphene-based nanomaterials [58,59], and zinc oxide nanoparticles [60], have been reported to enhance lysosomal acidity and ALP activity through the activation of transcription factor EB. Nevertheless, whether ATT functions via similar mechanisms remains unclear, and its precise reacidification mechanism warrants further investigation. Further elucidating the mechanism by which ATT restores lysosomal acidification would not only expand its therapeutic potential in diseases associated with lysosomal/autophagy–lysosomal system dysfunction but also facilitate the development of combination therapies with existing lysosomal acidifying agents (e.g., PLGA, PLA) [19,35,54,55]. By leveraging their complementary mechanisms of action, such synergistic interventions may enable dual-functional platforms combining pH-sensitive drug release with lysosomal repair and offer enhanced therapeutic efficacy for related disorders.

ATT demonstrates significant potential in clinical drug safety management and the treatment of lysosomal dysfunction-related diseases. It is indicated that ATT can effectively modulate the cytotoxicity of CQ. Similar to widely used antimalarial and autoimmune disease therapeutics, the long-term use of CQ may lead to severe adverse effects, including retinal damage and cardiotoxicity [61,62]. ATT could significantly reduce its toxicity in various cell types by inhibiting CQ-induced lysosomal alkalinization, offering a novel solution for clinical drug safety. Furthermore, ATT showed remarkable therapeutic effects on lysosomal dysfunction-related diseases, such as NAFLD. Experimental evidence confirmed that ATT could restore lysosomal acidity and autophagy flux in NAFLD models, effectively alleviating lipid accumulation. Recent studies have also identified independent anti-inflammatory properties of ATT [28,29], expanding its clinical application prospects. Chemotherapy-induced lysosomal dysfunction has emerged as a novel pathogenic mechanism contributing to cardiomyopathy, and restoring lysosomal acidification and function in cardiomyocytes may represent a promising therapeutic strategy [55]. Current clinical guidelines for NAFLD present limited pharmacological options, with only vitamin E and the PPAR-γ agonist pioglitazone being recommended by major hepatology associations for selected patient populations. Vitamin E demonstrates hepatoprotective efficacy through its antioxidant mechanisms, with randomized controlled trials confirming its ability to ameliorate hepatic steatosis and improve liver histology. Pioglitazone, while effective in enhancing insulin sensitivity and glycemic control, requires careful benefit–risk assessment due to its associated adverse effects. Future research directions include the following: (1) developing ATT as a toxicity-reducing synergist for clinical drugs and (2) exploring combination therapies with existing NAFLD medications. These studies will provide crucial scientific foundations for expanding ATT’s clinical applications. Lastly, while ATT has been approved as a food processing aid, due to its stability and safety, and demonstrates broad potential for biomedical applications, careful evaluation of its long-term use risks remains imperative.

## 5. Conclusions

Our study demonstrates that ATT nanorods function as a potent lysosomal modulator with broad therapeutic applications. By effectively restoring lysosomal acidity and autophagic flux, ATT not only counteracts chloroquine-induced cytotoxicity but also exhibits remarkable efficacy in diverse disease models, including NAFLD (via lipid droplet clearance) and Huntington’s disease (through enhanced mutant huntingtin degradation). Mechanistically, ATT promotes cathepsin D maturation and autolysosome formation, facilitating the clearance of pathogenic substrates. These findings position ATT as a promising therapeutic agent for lysosomal dysfunction-related disorders, spanning metabolic, neurodegenerative, and drug-induced pathological conditions.

## Figures and Tables

**Figure 1 biomolecules-15-00728-f001:**
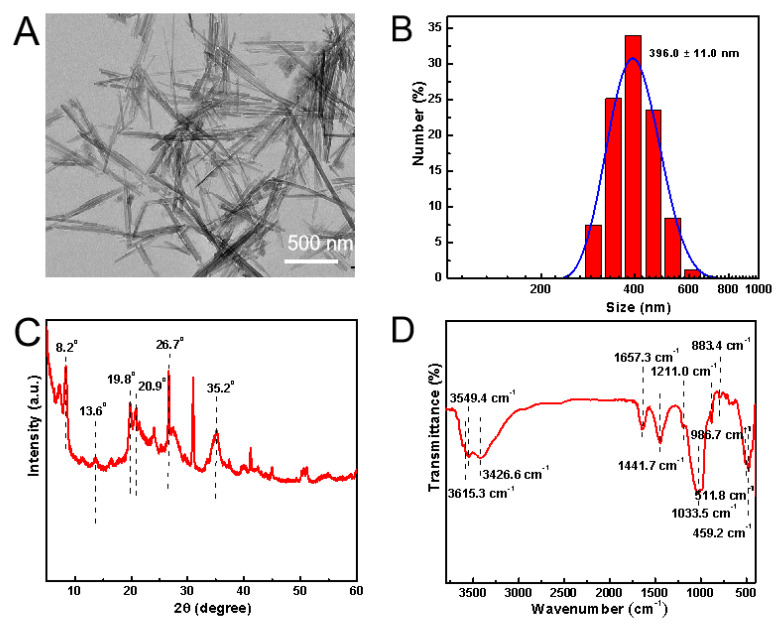
Characterizations of ATT. TEM image (**A**), hydrodynamic particle size (**B**), XRD pattern (**C**), and FTIR spectrum (**D**) of ATT.

**Figure 2 biomolecules-15-00728-f002:**
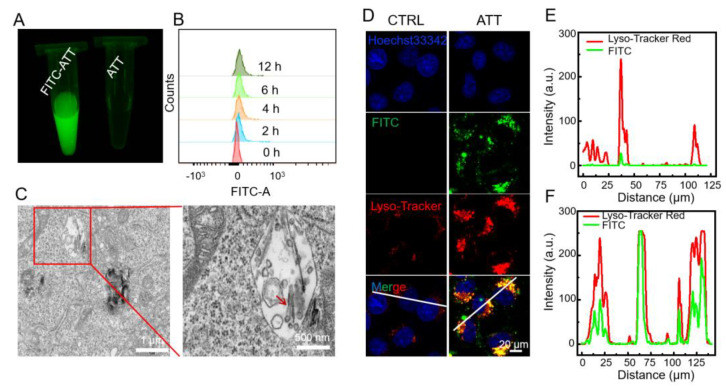
Cell uptake and intracellular localization of ATT. (**A**) FITC-labeled ATT nanorods confirmed by fluorescence imaging. (**B**) Flow cytometric analysis of time-dependent (0 h, 2 h, 4 h, 6 h, 12 h) internalization of FITC-labeled ATT in HeLa cells. (**C**) The morphology of ATT (as indicated by the arrow) observed under a transmission electron microscope JEM-1400 from JEOL in HeLa cells treated with ATT (500 μg/mL) for 6 h. Scale bar: 500 nm. (**D**) Representative images of fluorescence staining observed and captured by a laser scanning confocal microscope (Zeiss LSM880), showing the colocalization of FITC-labeled ATT and lysosomes labeled by Lyso-Tracker Red (75 nM; 30 min) in HeLa cells treated with ATT (500 μg/mL) for 6 h. Scale bar: 20 μm. (**E**,**F**) Line chart of fluorescence signal positioning analysis by ImageJ 1.46.

**Figure 3 biomolecules-15-00728-f003:**
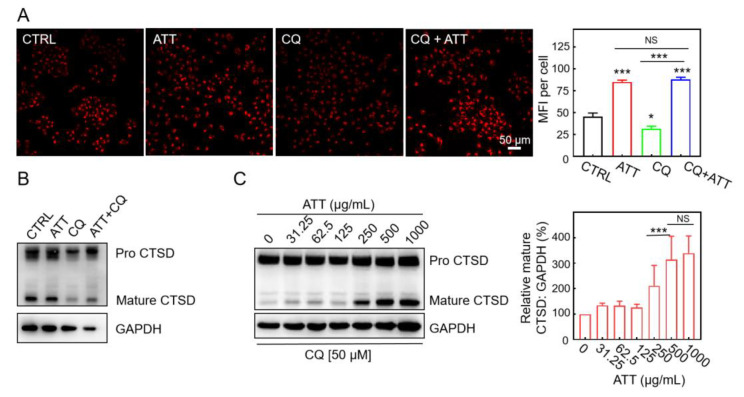
Restoration of lysosomal acidity and CTSD maturation induced by ATT in CQ-treated HeLa cells. (**A**) Fluorescence images of Lyso-Tracker Red (75 nM; 30 min) staining in HeLa cells treated with ATT (500 μg/mL) and CQ (50 μM) for 24 h. The mean fluorescence intensity (MFI) was analyzed by ImageJ 1.46. Over 50 cells were analyzed, n = 5. Scale bar: 50 μm. (**B**) Immuno-blotting analysis of cathepsin D in HeLa cells with the indicated treatments for 24 h. ATT, 500 μg/mL; CQ, 50 μM. (**C**) Immuno-blotting analysis of cathepsin D expression in HeLa cells treated with the different concentrations of ATT for 24 h in the presence of CQ. The experiment was repeated thrice and quantified by ImageJ 1.46 and GraphPad Prism 8 in the right panel. NS: no significant difference; * *p* < 0.05, *** *p* < 0.001. Original images of (**B**,**C**) can be found in Appendix A.

**Figure 4 biomolecules-15-00728-f004:**
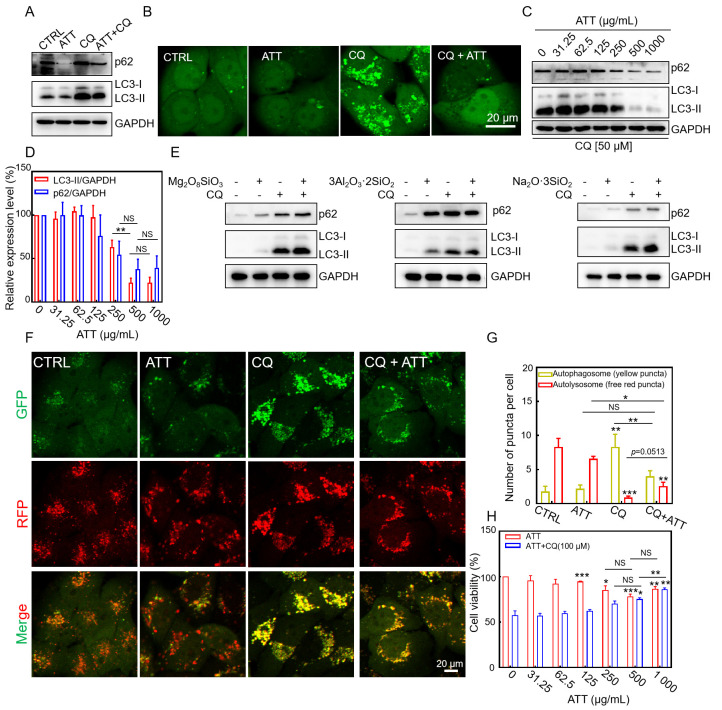
Restoring autophagic substrate protein degradation and autophagic flux by ATT nanorods in CQ-treated HeLa cells. (**A**) Immuno-blotting analysis of autophagic substrate expression in HeLa cells with the indicated treatments for 24 h. ATT (500 μg/mL); CQ (50 μM). (**B**) Fluorescence images of GFP-LC3/HeLa cells treated with ATT (500 μg/mL) and CQ (50 μM) for 24 h. Scale bar: 20 μm. (**C**) Immuno-blotting analysis of autophagic substrate expressions in HeLa cells treated with the different concentrations of ATT for 24 h in the presence of CQ (50 μM). (**D**) Three repeated analyses of Figure (**C**) were performed and analyzed by ImageJ 1.46 and Origin 2018. (**E**) Immuno-blotting analysis of autophagic substrate expressions in HeLa cells treated with the indicated treatments for 24 h. Mg_2_O_8_SiO_3_ (500 μg/mL), 3Al_2_O_3_·2SiO_2_ (500 μg/mL), Na_2_O_3_·SiO_2_ (500 μg/mL), and CQ (50 μM). (**F**) Fluorescent microscope images of mRFP-GFP-LC3 HeLa cells after the indicated treatments for 24 h. ATT (500 μg/mL); CQ (50 μM). Scale bar: 20 μm. (**G**) The quantitative analyses of Figure F were performed by ImageJ 1.46 and GraphPad Prism 8. (**H**) Cell viability analysis of HeLa cells treated with different doses of ATT in the presence or absence of CQ (100 μM) for 24 h. NS, no significant difference; * *p* < 0.05, ** *p* < 0.01, and *** *p* < 0.001. Original images of (**A**,**C,E**) can be found in Appendix A.

**Figure 5 biomolecules-15-00728-f005:**
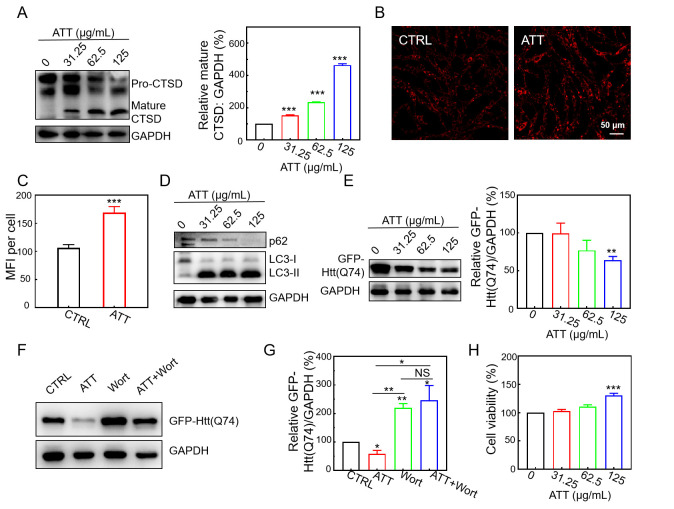
Increasing lysosomal acidity and accelerating the clearance of mutant huntingtin in GFP-Htt(Q74)/PC12 cells treated with ATT. (**A**) Immuno-blotting analysis and the quantitative analyses of CTSD in GFP-Htt(Q74)/PC12 cells treated with the indicated concentrations of ATT for 72 h. Three experiments were performed. (**B**,**C**) Fluorescence images of Lyso-Tracker Red (75 nM; 30 min) staining in GFP-Htt(Q74)/PC12 cells treated without or with ATT (125 μg/mL) for 72 h. The mean fluorescence intensity (MFI) was analyzed by ImageJ 1.46. Over 40 cells were analyzed, n = 5. Scale bar: 50 μm. (**D**) Immuno-blotting analysis of LC3 and p62 in GFP-Htt(Q74)/PC12 cells treated with the indicated concentrations of ATT for 72 h. (**E**) Immuno-blotting analysis and the quantitative analysis of mutant huntingtin in GFP-Htt(Q74)/PC12 cells treated with the indicated concentrations of ATT for 72 h. Three experiments were performed. (**F**,**G**) Immuno-blotting analysis and the quantitative analysis of mutant huntingtin expression in GFP-Htt(Q74)/PC12 cells treated with the indicated treatments. ATT (125 μg/mL), 72 h; wortmannin (Wort, 500 nM), 24 h. Three experiments were performed. (**H**) Cell viability analysis of GFP-Htt(Q74)/PC12 cells treated with the different doses of ATT for 72 h. NS: no significant difference; * *p* < 0.05, ** *p* < 0.01, *** *p* < 0.001. Original images of (**A,D,E,F**) can be found in Appendix A.

**Figure 6 biomolecules-15-00728-f006:**
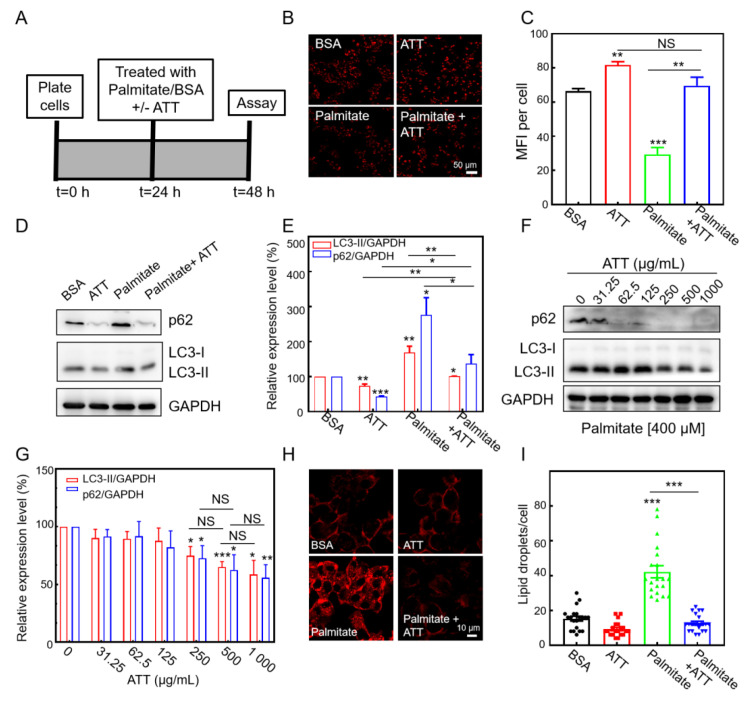
ATT reacidified HepG2 cells and restored autophagy flux under lipotoxicity. (**A**) A schematic of the experimental protocol for cell treatment for 24 h before assaying for lysosomal acidity, autophagy flux, or lipid droplet accumulation. (**B**,**C**) Lyso-Tracker Red (75 nM; 30 min) staining (**B**) and the corresponding quantitative (**C**) analysis of HepG2 cells exposed to different treatments for 24 h. PA, 400 μM; ATT, 250 μg/mL. The quantitative analyses were performed by ImageJ 1.46 and GraphPad Prism 8. n = 3. Scale bar: 50 μm. (**D**,**E**) Immuno-blotting (**D**) and the corresponding quantitative (**E**) analysis of autophagy substrate expressions in HepG2 cells exposed to different treatments for 24 h. PA, 400 μM; ATT, 250 μg/mL. Three experiments were performed. (**F**,**G**) Immuno-blotting (**F**) and the corresponding quantitative analysis (**G**) of autophagic substrate expressions in HepG2 cells treated with different concentrations of ATT for 24 h in the presence of PA (400 μM). Four individual experiments were performed. (**H**) Representative confocal images of HepG2 cells stained with Nile red dye (0.5 μg/mL) for 15 min and imaged by fluorescence microscopy. The Nile red dye accumulates rapidly in lipid vesicles. Scale bar: 10 μm. (**I**) Quantification of lipid vesicle number indicating a significant reduction in lipid droplet density after ATT treatment in HepG2 cells exposed to palmitate. Control ATT addition did not reduce lipid droplets (n = 20 cells analyzed per condition). HepG2 cells exposed to different treatments for 24 h. PA, 400 μM; ATT, 250 μg/mL. The quantitative analyses were performed by ImageJ 1.46 and GraphPad Prism 8. NS: no significant difference; * *p* < 0.05, ** *p* < 0.01, *** *p* < 0.001. Original images of (**D,F**) can be found in Appendix A.

**Figure 7 biomolecules-15-00728-f007:**
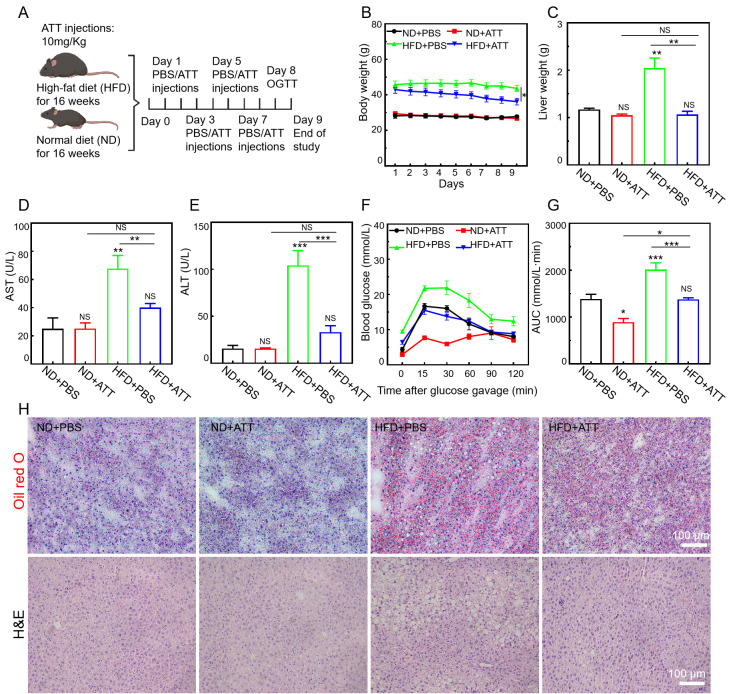
Reversal of HFD-induced NAFLD with ATT. (**A**) In the animal trials, mice were fed either a normal diet (ND) or an HFD for 16 weeks. And then, every other day, mice from different groups were intravenously tail-vein-injected with ATT (10 mg/kg) or PBS four times. (**B**,**C**) Body weight and liver weight of mice under different treatments. (**D**,**E**) Serum AST level and serum ALT level of mice under different treatments. (**F**,**G**) The oral glucose tolerance test (OGTT) was carried out on mice that were fasted overnight for 12 h. After determination of fasted blood glucose levels, each mouse was administered 2 g/kg body weight of glucose by intragastric administration. The blood glucose level was detected from the tail vein after 15, 30, 60, 90, and 120 min. (**H**) Oil Red O and H&E staining of mouse livers. n ≥ 3. Scale bar: 100 μm. NS: no significant difference; * *p* < 0.05, ** *p* < 0.01, and *** *p* < 0.001.

**Figure 8 biomolecules-15-00728-f008:**
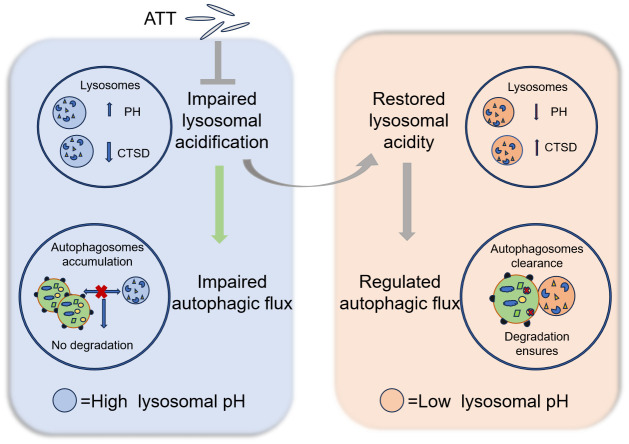
Pattern diagram for rescuing autophagy flux and lysosomal dysfunction by ATT. Created by PowerPoint.

## Data Availability

The original contributions presented in this study are included in the article/Appendix A. Further inquiries can be directed to the corresponding author(s).

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
