# Peer review of "Recovery of Lysosomal Acidification and Autophagy Flux by Attapulgite Nanorods: Therapeutic Potential for Lysosomal Disorders"

_biomolecules, 2025, doi:10.3390/biom15050728_

Round 1

Reviewer 1 Report

Comments and Suggestions for Authors

-In this study, the authors investigated the therapeutic potential of attapulgite (ATT) nanorods in restoring lysosomal acidification and autophagic flux particularly in disease contexts where these pathways are impaired. They have shown that ATT can localize to lysosomes, reacidify their environment and promote the maturation of a key lysosomal enzyme called cathepsin D. Utilizing both in vitro and in vivo models, such as chloroquine-treated cells, Huntington’s disease cell lines, and non-alcoholic fatty liver disease (NAFLD) mice, they revealed that ATT alleviates autophagy blockage, reduces toxic protein and lipid accumulation and improves cellular and metabolic health.

-The manuscript would benefit from a schematic or graphical figure that illustrates the context of lysosomal-based disorders and the proposed mechanism of action of attapulgite nanorods.

-Lines 37, 42, 51, References should be placed at the end of the sentence with one blank space on each side. Please revise the entire manuscript accordingly.

-Line 40, Abbreviations should be introduced immediately after the full term is defined, not afterwards. Please revise it.

-There are several instances of unnecessary blank spaces (e.g., Line 45, 79) within the manuscript. Please remove all extra spacing.

-Line 73, Since chloroquine has already been defined as “CQ” in the abstract, there is no need to redefine it again in the main text.

-Line 488, Latin terms such as “in vitro” should be italicized consistently throughout the manuscript.

-Lines 476-482, The paragraph outlining future research directions for ATT could be enriched by discussing previous studies and pointing out its potential integration into current biomedical applications, such as drug delivery platforms, nanocarriers, or co-treatment strategies with approved therapeutics.

-The “Materials and Methods” section (Section 4) should be relocated to follow the Introduction to align with conventional formatting standards.

Reviewer 2 Report

Comments and Suggestions for Authors

Dear authors, this is a certainly interesting paper demonstrating high quality and novelty. Please find attach my kind suggestions in order to improve your manuscript.

Reviewer 3 Report

Comments and Suggestions for Authors

The manuscript titled “Recovery of Lysosomal Acidification and Autophagy Flux by Attapulgite Nanorods: Therapeutic Potential for Lysosomal Disorders” by Fan, X.; et al. is a scientific work where the authors addressed the positive effect of attapulgite to maintain the homeostasis of lysosomal metabolism. The most relevant outcomes found in this research could serve to pave the way in the design of the next-generation of smart therapies against non-alcoholic liver diseases and neurodegenerative malignancies. The manuscript is generally well-written and this is a topic of growing interest.

However, it exists some points that need to be addressed (please, see them below detailed point-by-point) to improve the scientific quality of the submitted manuscript paper before this article will be consider for its publication in Biomolecules.

1) Keywords. The authors should consider to add the term “neurodegenerative malignancies” in the keyword list.

2) Introduction. “Dysfunction of the lysosome and autophagy-lysosome system is closely associated with various diseases, such as neurodegenerative diseases (…) Huntington’s disease (HD) is closely related to dysfunction of the authophagy-lysosome pathway (ALP)” (lines 55-64). Could the authors provide quantitative data insights according to the worldwide global burdens of incidence of those human diseases related to lysosome dysfunction processes? This will significantly aid the potentail readers to better understand the significance of this devoted research.

3) Results. “Measurements conducted using Malvin’s nanoparticle size analyzer (…) 396.00 ± 11.00 nm” (lines 115-117). Is the sensitivity of this technique below the Ångstrom?

4) “Furthermore, bio-TEM imaging analysis (…) cellular internalization” (lines 139-141). Would the authors mean “cryo-TEM” instead of “bio-TEM”? Some insights should be furnished in this regard.

5) “2.2. Cellular uptake and intracellular localization of ATT” (lines 134-159). Did the authors carry out fluorescence lifetime imaging microscopy (FLIM) measurements to probe the exerted dynamic molecular environment interactions during the cellular uptake of attapulgite?

6) “2.4. Restoring CQ-blocked flux and reducing CQ-induced cytotoxicity by ATT”. “Autophagic flux refers to the complete dynamic process (…) formation of autophagosomes, the fusion of autophagosomes with lysosomes, and the degradaton and recycling of autophagic substrates” (lines 198-202). Here, even if I agree with the information provided in this statement, it should be opportune to go further in the underlying mechanisms of autophagic lysosome reformation [1] and the pivotal role of clathrin lattices [2].

[1] https://doi.org/10.1038/ncb2557

[2] https://doi.org/10.1101/2024.10.01.616068

7) Finally, did the authors carry out experiments in presence of ATP or ADP (as negative control) to check how this energetic metabolic molecules can impact on the lysosomal homeostasis and subsequent autophagy processes?

8) “3. Discussion” (lines 445-516). This section perfectly remarks the most relevant outcomes found by the authors in this work and also the promising future prospectives. It may be advisable to add a brief statement to remark the potential future action lines to pursue the topic covered in this work.

9) Materials & Methods. “4.7. Observation of ATT by TEM” (lines 576-583). What is the electron acceleration voltage rate to gather the TEM images?

Reviewer 4 Report

Comments and Suggestions for Authors

Fan et al. investigate the therapeutic potential of attapulgite (ATT), a natural nanomaterial, in enhancing mutant huntingtin protein clearance in Huntington’s disease models and improving lipid metabolism and glucose regulation in NAFLD mice. While the use of ATT as a lysosomal modulator is intriguing, several important issues need to be addressed:

  1. The rationale for selecting Huntington’s disease and NAFLD as models of lysosomal dysfunction is unclear. The manuscript lacks a unifying theme to justify focusing on only these two conditions. If the intent is to broadly test ATT across lysosome-related disorders, why were additional models not included for comparison?
  2. In the introduction, CQ is a known lysosomotropic agent that raises lysosomal pH to treat autoimmine diseases. In this paper, the authors focus on lowering lysosomal pH. Hence, there is no basis of comparison to CQ for its therapeutic effects, since they are just two contrasting agents. The authors should clarify this rationale and also review and discuss other relevant agents or materials that have been developed to acidify lysosomes in the discussion session.
  3. The authors should include data on whether ATT treatment increases lysosomal acidity (Lysotracker staining) in the GFP-Htt(Q74)/PC12 cells, which would further support its proposed mechanism of action.
  4. In Figure 6D, the p62 band for the palmitate + ATT condition appears blank (white), yet the quantification shown in Figure 6E is high. Please provide an explanation or clarify whether this discrepancy results from image quality or experimental variation.
  5. Statistical analysis is missing for Figure 6G and should be included to support the significance of the findings.
  6. Please provide justification for the timing and dosage of ATT treatment in HFD mice - what criteria were used to select these specific timepoints and concentrations?

Round 2

Reviewer 1 Report

Comments and Suggestions for Authors

The authors responded to comments requested during the revision of the article.

Reviewer 4 Report

Comments and Suggestions for Authors

The authors have addressed my comments.